Telehealth satisfaction among patients with chronic diseases: a cross-sectional analysis

Hendy Abdelaziz abdelaziz.hendy@nursing.asu.edu.eg 1
Farghaly Abdelaliem Sally Mohammed 2
Zaher Ahmed 3
Sadek Bothayna N. 4
Nashwan Abdulqadir J. 5
Al-Jabri Mohammed Musaed Ahmed 6
Ahmeda Ahmad 7 8
Hendy Ahmed 9 10
Alabdullah Amany Anwar Saeed 12
Sinnokrot Shaban Majed 11
1 Pediatric Nursing Department, Faculty Nursing Ain Shams University , Cairo , Egypt
2 Department of Nursing Management and Education, College of Nursing, Princess Nourah bint Abdulrahman University , Riyadh , Saudi Arabia
3 Psychiatric Mental Health Nursing, Faculty of Nursing, Ain Shams University , Cairo , Egypt
4 Assistant Professor, Pediatric Nursing Department, Faculty Nursing Ain Shams University , Cairo , Egypt
5 Department of Public Health, College of Health Sciences, QU Health, Qatar University , Doha , Qatar
6 College of Applied Medical Sciences, Prince Sultan University , Wadi Aldawaser , Saudi Arabia
7 Department of Basic Medical Sciences, College of Medicine, Ajman University , Ajman , United Arab Emirates
8 Centre of Medical and Bio-allied Health Sciences Research, Ajman University , Ajman , United Arab Emirates
9 Department of Mechanics and Mathematics, Western Caspian University , Baku , Azerbaijan
10 Department of Computational Mathematics and Computer Science, Institute of Natural Sciences and Mathematics, Ural Federal University , Yekaterinburg , Russia
11 Faculty Nursing, Zarqa University , Zarqa , Jordan
12 Department of Maternity and Pediatric Nursing, College of Nursing, Princess Nourah bint Abdulrahman University , Riyadh , Saudi Arabia
Moran Jose
Electronic publication date: 2025 Apr 25
Publication date: 2025
Volume: 13
Electronic Location ID: e19245
Received 2024 Jul 30; Accepted 2025 Mar 11
Copyright: ©2025 Hendy et al.
Copyright year: 2025
Copyright holder: Hendy et al.
License: This is an open access article distributed under the terms of the Creative Commons Attribution License, which permits unrestricted use, distribution, reproduction and adaptation in any medium and for any purpose provided that it is properly attributed. For attribution, the original author(s), title, publication source (PeerJ) and either DOI or URL of the article must be cited.
License URL: https://creativecommons.org/licenses/by/4.0/

Keywords: Telehealth utilization, Chronic diseases, Telehealth services, Patient satisfaction, Logistic models

Funding: Princess Nourah bint Abdulrahman University Researchers Supporting Project PNURSP2025R444 Princess Nourah bint Abdulrahman University Researchers Supporting Project number (PNURSP2025R444), Princess Nourah bint Abdulrahman University, Riyadh, Saudi Arabia. The funders had no role in study design, data collection and analysis, decision to publish, or preparation of the manuscript.

==============================
Background

The study aims to assess telehealth satisfaction among patients with chronic diseases focusing on key demographic and clinical factors that influence satisfaction.

Methods

A descriptive cross-sectional study was conducted using a self-reported online questionnaire between December 1, 2023, and January 30, 2024. The study targeted chronic patients who had been using telehealth for at least three months. After screening for eligibility and ensuring data completeness, responses from 1,070 patients from three non-governmental hospitals were included in the analysis. The questionnaire covered demographic, socio-economic, and technology-related data, as well as a telehealth satisfaction scale.

Results

A total of 62.9% of patients reported being satisfied with the telehealth services they received, while 37.1% expressed dissatisfaction. Logistic regression analysis identified several factors associated with patient satisfaction. The constant term was significantly positive (coefficient = 4.129, p < 0.001), indicating a baseline high level of satisfaction. Age negatively impacted satisfaction (coefficient = −0.191, p < 0.001), with older patients being less satisfied. Male patients showed a higher satisfaction rate (coefficient = 0.473, p = 0.047), while education level, particularly having a bachelor’s degree, was strongly associated with increased satisfaction (coefficient = 1.977, p < 0.001). Marital status (married) was not a significant predictor (p = 0.403), whereas employment status (working) had a positive association with satisfaction (coefficient = 1.445, p < 0.001). Income level (sufficient and save) did not significantly affect satisfaction (p = 0.561). Having children was positively associated with satisfaction (coefficient = 1.189, p < 0.001).

Conclusion

Addressing demographic, socio-economic, and technological needs can enhance patient satisfaction with telehealth services. Tailoring services to specific patient preferences, especially for older patients and those needing continuous training, can improve telehealth effectiveness and acceptance.

Introduction

A chronic disease is a long-lasting condition that persists for a year or more and requires ongoing medical attention or limits activities of daily living, or both (Wang et al., 2016). Chronic diseases have consistently posed a major challenge in health management and public health, with the Global Burden of Disease 2015 study reporting that they accounted for over 50% of global deaths. Among these diseases, cardiovascular diseases, diabetes, and chronic obstructive pulmonary disease (COPD) were identified as the three main sources requiring continuous monitoring, management, and patient education (Wang et al., 2016). Hence, the development of chronic disease management systems is crucial, with efficiency and effectiveness being the two primary concerns (Xiao & Han, 2023). In recent years, telehealth has rapidly developed to realize more efficient chronic health management, aiming to achieve more efficient management of chronic health conditions (Amjad, Kordel & Fernandes, 2023). Telehealth provides an effective means to remotely monitor patients’ vital signs, deliver personalized care plans, and empower patients to actively engage in self-management, enhancing disease management, increasing patient adherence to treatment plans, and reducing the frequency of hospital admissions (Creber et al., 2023).

Telehealth, encompassing telemedicine and telecare, utilizes technology to facilitate remote healthcare delivery and overcome barriers of distance and time, enabling healthcare providers to reach patients beyond traditional clinical settings, improving access to care and reducing the burden on healthcare systems (Babu, Sudha & Caroline Jebakumari, 2023). Telehealth can be broken down into several key components, including telemedicine, which enables remote consultations between patients and healthcare providers; remote patient monitoring (RPM), which allows continuous tracking of patient health data via wearable devices or sensors; and mobile health (mHealth) applications, which provide patients with tools to manage their health on a day-to-day basis. These approaches are particularly valuable in addressing the needs of underserved populations, improving care continuity, and enhancing chronic disease management (Creber et al., 2023). The roots of telehealth can be traced back to the 1960s when early telemedicine programs were developed. In 1964, the National Aeronautics and Space Administration (NASA) conducted experiments using telemedicine to monitor astronauts’ health during space missions (Amjad, Kordel & Fernandes, 2023; Babu, Sudha & Caroline Jebakumari, 2023). Telehealth offers a wide range of services, including remote consultations, monitoring, education, and support, making it an invaluable tool in managing chronic diseases (Mathew et al., 2023). Compared to community care without digital tools, telehealth can significantly reduce the time and costs associated with health status monitoring (Sayani et al., 2019). There are also fewer requirements for labor and physical presence, as care leaders and offline participation are no longer indispensable (Haleem et al., 2021). Additional benefits include decreased hospitalization rates and decreased manual errors in data entry, as mobile devices can record medical data frequently and remotely (Gajarawala & Pelkowski, 2021). Recent studies have shown that telehealth can lead to improved clinical outcomes, increased patient satisfaction, and cost savings for both patients and providers (Xiao & Han, 2023; Creber et al., 2023).

In Egypt, telehealth services are gradually gaining traction, driven due to the increasing availability of technology infrastructure and the recognition of its potential to address healthcare challenges with improved technology infrastructure, including internet connectivity and mobile devices, telehealth is becoming more accessible to a larger population (Osman et al., 2023). While minor strides have been made in the field of telehealth, more progress is still needed, particularly in the context of achieving universal health coverage (Shouman et al., 2021). Leading to expanded opportunities for remote healthcare delivery accelerated by the COVID-19 pandemic and minimizing in-person contact, telehealth also helped to alleviate patients’ depression and anxiety by providing remote treatment and consultation channels when offline hospital visits were impeded (Liu et al., 2020). It seems that telehealth was not just an efficient system but also effective, as indicated by a study on patients’ perception regarding telemonitoring. Nearly 90% of patients reported feeling satisfied with the care and becoming more knowledgeable about their disease (Li et al., 2021).

Patient satisfaction is essential for the successful implementation of telehealth and for guiding action plans aimed at improving the quality of telehealth services (Kruse et al., 2017). Patient satisfaction refers to the extent to which patients’ expectations and needs are met during their healthcare experience, encompassing several dimensions. These dimensions include the quality of healthcare services received, communication with healthcare providers, accessibility, convenience, and the overall patient experience (Ferreira et al., 2023). It has been observed that telehealth is associated with a higher level of satisfaction among both patients and providers (Mason, 2022). Consequently, high levels of patient satisfaction contribute to increased patient engagement, improved adherence to treatment plans, and positive word-of-mouth recommendations, thereby driving the utilization and sustainability of telehealth services. Conversely, low patient satisfaction may result in disengagement, discontinuation of services, and limited utilization of telehealth, undermining the potential benefits it offers (Hailu et al., 2024).

Several factors influence patient satisfaction in telehealth, including effective communication between healthcare providers and patients, personalized care, accessibility of telehealth services, ease of use of technology, availability of support and education, privacy and security measures, and a patient-centered approach (Pogorzelska & Chlabicz, 2022). Furthermore, studies have found that the quality of telehealth services and the clinical outcomes of patients following telehealth visits may be comparable to those of traditional face-to-face office visits, with the added benefit of improved access to care (Hailu et al., 2024; Phillips et al., 2023; Chen, Lodaria & Jackson, 2022). Therefore, the purpose of this study was to comprehensively examine the effect of telehealth on the satisfaction of patients with chronic diseases.

Aim

The aim of this study is to assess telehealth satisfaction among patients with chronic diseases focusing on key demographic and clinical factors that influence satisfaction.

Objectives

-To assess the overall level of patient satisfaction with telehealth services.

-To analyze the impact of demographic and socio-economic factors on patient satisfaction with telehealth.

Methods

Study design

A descriptive cross-sectional study using a self-report online questionnaire was conducted and reported in accordance with the guidelines for Strengthening the Reporting of Observational Studies in Epidemiology (STROBE). The online questionnaire was distributed to patients from December 1, 2023, to January 30, 2024.

The cross-sectional approach was selected as it allows for the collection of data from participants during one time frame without follow-up or longitudinal tracking. This design is particularly useful for obtaining a snapshot of telehealth utilization and its associated outcomes in chronic disease management.

Sampling and recruitment

A purposive sample of patients with chronic diseases who used telehealth for follow-up consultations with their physicians. The inclusion criteria required that patients be able to read and write, have used telehealth for at least three months, suffer from a chronic disease, and be willing to participate in the study. Both female and male patients were included. However, to improve generalizability, future studies should consider including participants with varying literacy levels and first-time users of telehealth to capture a broader range of experiences. Additionally, expanding the sample to include patients from public healthcare settings and rural areas would ensure a more comprehensive representation of the usual care population, reflecting diverse socioeconomic and geographical contexts.

Sample size

The method for estimating the sample size in logistic regression is the rule of thumb proposed by Peduzzi et al. (1996), which suggests that the minimum number of events per predictor should be at least 10. The formula is:

n = 10k /p(1−p)

where:

• n is the minimum sample size.

• k is the number of predictors = 12

• p is the proportion of cases in the population associated with the key outcome, which we set at 0.62 based on satisfaction rates reported in similar studies (e.g., Gado, El Salamony & Zewiel, 2022) where 62% of patients were satisfied with telepsychiatry.

n = 10∗12/0.62∗(1 − 0.62)

n = 120/(0.62∗0.38)

n = 120/0.2356 = 509

Using this formula, the minimum sample size required for the analysis was approximately 509 participants to ensure sufficient power for detecting significant associations. This calculation was based on achieving a balance between statistical rigor and practical feasibility, considering the number of predictors included in the model. Ultimately, data were collected from 1070 eligible participants, exceeding the minimum requirement and further enhancing the robustness of the results.

Setting

The study was conducted in three non-governmental hospitals in Cairo Governorate, Egypt, where telehealth services—primarily teleconsultation (remote video or audio-based consultations between patients and healthcare providers) and mobile health (patient education and follow-up care via mobile applications and SMS services)—were provided for chronic patients as an alternative to traditional outpatient follow-ups, particularly routine follow-ups, from December 2023 to January 30, 2024. A total of 1,915 patients with chronic diseases were invited to participate after confirming their eligibility based on the previously stated criteria. Of these, 419 patients (21.8%) declined to participate, and 191 patients withdrew during data collection, resulting in an initial dataset of 1,305 participants. Following data cleaning and revision, 235 entries were excluded due to missing or incomplete data, yielding a final analyzed sample size of 1,070 patients (Fig. 1).

Figure 1 Flowchart of study.

A larger sample helps reduce variability and enhances the reliability of statistical estimates, particularly when using logistic regression models. More recent studies have suggested that, in some cases, increasing the sample size beyond this threshold further improves precision, particularly when analyzing complex interactions and subgroup differences (Riley et al., 2019). Additionally, a larger sample size helps mitigate type II errors, ensuring sufficient power to detect true associations between independent variables and the outcome variable (Van Smeden et al., 2019). Also, Similar studies on telehealth satisfaction among chronic patients have used large sample sizes to ensure meaningful subgroup analysis and robust statistical conclusions (e.g., Polinski et al., 2016; Blavin, 2023).

Tools of data collection

Data were collected using a self-administered online questionnaire. This questionnaire was developed based on an extensive review of relevant study and previously published instrument (Gado, El Salamony & Zewiel, 2022). The questionnaire, originally in English, was translated into Arabic to ensure its applicability for Arabic-speaking populations. The scale was translated from the original language to Arabic using the forward-backward translation method to ensure conceptual equivalence. Two bilingual native Arabic translators initially translated the scale, followed by a back-translation into the original language by a third bilingual translator. Discrepancies were resolved through team discussions. To ensure cultural relevance, three subject matter experts with expertise in both the content and cultural context of the Arab society reviewed the translated items.

Part I

This part included demographic and socioeconomic factors, as well as aspects related to telehealth use, such as age, sex, education level, marital status, employment status, income level, have children, use telehealth again, recommend telehealth to others, easy to contact with it at hospital, residence, continuous training about using telehealth.

Part II: Telehealth satisfaction scale

This tool, developed by Morgan et al. (2014), was adopted and translated into Arabic. It is used to assess patients’ satisfaction levels with telehealth services and includes 10 items that evaluate thoroughness, carefulness, and skillfulness of the telehealth clinic team, explanation of treatment, and length of time with the telehealth team, among others. Scale items are rated on a 4-point Likert scale (1 = poor, 2 = fair, 3 = good, and 4 = excellent). The total score on the 10-item Telehealth satisfaction scale (TeSS) ranges from 10 to 40, with higher scores indicating greater satisfaction. A score between 28 and 40 is categorized as satisfaction, while a score between 10 and 27 is categorized as dissatisfaction.

Data collection process

We distributed an online self-administered questionnaire across Email, Facebook, WhatsApp, and other social media platforms for patients. The questionnaire consisted of three sections. The first section outlined the study objectives and eligibility criteria. The second section gathered demographic, socioeconomic, and technology-related information, as well as the Telehealth Satisfaction Scale. The online questionnaire was created using Google Forms, and the link was shared with patients.

Data collection spanned over two months. Participants had to provide online informed consent by clicking an “I agree” button before accessing the survey questions. The researchers closed the survey once the estimated sample size was reached. Finally, the data collected was revised for missing data, and any incomplete questionnaires were excluded. The complete questionnaires were entered into the Statistical Package for Social Sciences (SPSS) to avoid potential data entry errors.

Validity & reliability

The researcher ensured that the questionnaire content and the alignment of the questions effectively addressed the aims, objectives, and research questions of this study. Content validity was assessed using the Content Validity Index (CVI), based on expert evaluation. Five experts—professors specializing in public health, health informatics, and nursing—provided ratings on the relevance and clarity of each item. The overall content validity index exceeded the accepted threshold of 0.80, indicating satisfactory content validity.

The reliability of the tool was measured using Cronbach’s alpha test, which yielded the following result: TsSS (α = 0.837), demonstrating good reliability.

Factor analysis suitability was assessed using the Kaiser–Meyer–Olkin (KMO) test, which resulted in a value of 0.508, indicating moderate adequacy for factor analysis. Bartlett’s Test of Sphericity produced a chi-square value of 2,781.94 (p < 0.001), confirming significant correlations among variables and supporting the scale’s factorability.

Confirmatory factor analysis (CFA) verified the factor structure, with a Cronbach’s alpha (α) of 0.837, indicating good internal consistency and high reliability. Variance Inflation Factor (VIF) analysis showed that all items had values between 2 and 5, suggesting moderate correlation without multicollinearity concerns.

A pilot study involving 107 patients (10% of the total sample size) was conducted to assess the clarity, applicability, relevance, and feasibility of the survey tools and to estimate the data collection duration. No modifications were made based on the pilot study results, and the patients involved in the pilot were included in the main study sample.

Ethical consideration

This study was performed in accordance with the principles of the Helsinki Declaration and the relevant guidelines and regulations. Ethical approval to conduct the research was obtained from the Research Ethical Committee at the Faculty of Medicine, MTI University, ID 1023-2024. All participating patients were provided with comprehensive information about the study’s purpose, objectives, and potential benefits. Online informed consent was obtained from each patient prior to their participation. The researchers emphasized the voluntary nature of the study, and nurses had the option to withdraw their participation at any time without facing any consequences. To maintain confidentiality, the collected data was coded, ensuring that no identifiable information was disclosed

Statistical analysis

The data were analyzed using SPSS version 22.0 (IBM Corp., Armonk, NY, USA) and JASP. General characteristics were described using descriptive statistics. The logistic regression analysis identified several significant predictors for the dependent variable “Total satisfaction”. A logistic regression model was used to examine the relationship between “Total satisfaction” and a set of independent variables as age, sex (male = 1), education level (bachelor = 1), marital status (married = 1), employment status (work = 1), income level (sufficient and save = 1), have children (yes = 1), use telehealth again (yes = 1), recommend telehealth to others (yes = 1), easy to contact with IT at hospital (always = 1), residence (far from hospital = 1), and continuous training about using telehealth (yes = 1).

The receiver operating characteristic (ROC) curve generated for your binary logistic regression model demonstrates the model’s ability to distinguish between the two classes (“Satisfaction” vs. “Unsatisfaction”). Area under curve (AUC) ranges from 0.5 (random model) to 1 (perfect model). Higher AUC values indicate better model performance. Statistical significance was set at P < 0.05. We checked the model accuracy and performance metrics: accuracy: 0.87, precision: 0.86, recall: 0.86, F1-score: 0.86. The model showed a high level of accuracy and good performance metrics, indicating it effectively predicts overall satisfaction with telehealth services.

Results

Table 1 presents a comprehensive demographic profile of the 1,217 studied participants. The data reveals a diverse age distribution, with a mean age of 50.27 years (SD = 14.8) and an age range from 25 to 75 years. Sex-wise, the sample is relatively balanced, with 48.2% male and 51.8% female participants. The educational qualifications vary, with the highest percentage (65.2%) holding a bachelor’s degree, followed by 18.3% with preparatory education and 16.5% with secondary education. The majority of participants (77.3%) are married, while 22.7% are unmarried. In terms of income, 49.1% of participants report having sufficient income, 36.4% have sufficient income and can save, and 14.5% find their income insufficient. A significant majority of participants (82.3%) have children, while 17.7% do not. Most participants (67.6%) always find it easy to contact IT at the hospital, 20.9% sometimes find it easy, and 11.5% never find it easy. Regarding continuous training about using telehealth, 71.3% have received training, while 28.7% have not. Most participants (85.2%) live far from the hospital, while 14.8% live near the hospital. The employment status is fairly balanced, with 52.4% of participants working and 47.6% not working.

Table 1 Characteristics of studied chronic disease patients (n = 1,070).

	n	%	
Age:			
Mean (SD)      50.27 (14.8)			
Range      25–75			
Gender			
Male	516	48.2	
Female	554	51.8	
Education level:			
Preparatory	196	18.3	
Secondary	176	16.5	
Bachelor	698	65.2	
Marital status:			
Married	827	77.3	
Unmarried	243	22.7	
Income level:			
Insufficient	155	14.5	
Sufficient	525	49.1	
Sufficient and save	390	36.4	
Having children:			
Yes	881	82.3	
No	189	17.7	
Easy to contact with IT at hospital			
Always	723	67.6	
Sometimes	224	20.9	
Never	123	11.5	
Continuous training about using tele-health			
Yes	763	71.3	
No	307	28.7	
Residence			
Far from hospital	912	85.2	
Near to hospital	158	14.8	
Employment:			
Work	561	52.4	
Not-work	509	47.6	

Figure 2 presents two bar charts illustrating patient responses regarding their telehealth experiences. The left chart shows that 84.9% of patients would use telehealth again, while 15.1% would not. The right chart indicates that 73.3% of patients would recommend telehealth to others, compared to 26.7% who would not. These results suggest high overall satisfaction and willingness to endorse telehealth services.

Figure 2 Patient engagement with telehealth services (n = 1, 070).

Figure 3 illustrates the overall patient satisfaction with telehealth services. The majority of patients, 62.9%, reported being satisfied with the telehealth services they received, represented by the blue bar. On the other hand, 37.1% of patients expressed dissatisfaction.

Figure 3 Overall patient satisfaction with telehealth services (n = 1, 070).

Table 2 presents the logistic regression results for factors associated with chronic patient satisfaction related to using telehealth services. The constant term is significantly positive (coefficient = 4.819, p < 0.001), indicating a baseline high satisfaction level. Age negatively impacts satisfaction (coefficient = −0.185, p < 0.001), suggesting that older patients are less satisfied with telehealth services. Gender (male) shows a positive association (coefficient = 0.581, p = 0.012), indicating higher satisfaction among male patients. Education level, specifically having a bachelor’s degree, strongly correlates with increased satisfaction (coefficient = 2.058, p < 0.001). Marital status (married) does not significantly impact satisfaction (p = 0.494). Employment status (working) is positively associated with satisfaction (coefficient = 1.500, p < 0.001). Income level (sufficient and save) is not significantly associated with satisfaction (p = 0.553). Having children is positively associated with satisfaction (coefficient = 1.155, p < 0.001). Ease of contact with IT at the hospital is a significant positive predictor (coefficient = 0.898, p < 0.001). Residence far from the hospital is also positively associated with satisfaction (coefficient = 1.048, p = 0.001). Continuous training about using telehealth is a strong positive predictor (coefficient = 1.500, p < 0.001).

Table 2 Logistic regression for satisfaction related using tele-health.

Feature	Coefficient	Standard error	z-value	p-value	Conf. Interval lower	Conf. Interval upper	
Constant	4.81	0.63	7.59	0.000	3.57	6.06	
Age	−0.18	0.01	−14.66	0.000	−0.20	−0.15	
Gender	0.58	0.23	2.51	0.012	0.12	1.03	
Education level (Bachelor)	2.05	0.25	8.04	0.000	1.55	2.55	
Marital status (Married)	−0.18	0.27	−0.68	0.494	−0.72	0.35	
Employment status (Work)	1.50	0.25	5.97	0.000	1.00	1.99	
Income level (Sufficient and save)	0.14	0.23	0.59	0.553	−0.32	0.60	
Have children (Yes)	1.15	0.29	3.87	0.000	0.57	1.73	
Easy to contact with IT at hospital (Always)	0.89	0.24	3.61	0.000	0.41	1.38	
Residence (Far from hospital)	1.04	0.31	3.30	0.001	0.42	1.67	
Continuous training about using telehealth (Yes)	1.49	0.26	5.72	0.000	0.98	2.01	
Notes.

R2 = 0.6237.

p-value < 0.001.

The receiver operating characteristic (ROC) curve in Fig. 4 evaluates the performance of the logistic regression model used to predict patient satisfaction with telehealth services. The curve plots the true positive rate (sensitivity) against the false positive rate (1-specificity) at various threshold settings. The area under the ROC curve (AUC) is 0.96, indicating excellent predictive performance. An AUC of 1.0 represents a perfect model, while an AUC of 0.5 suggests a model with no discriminative power. Therefore, an AUC of 0.96 demonstrates that the logistic regression model is highly effective in distinguishing between satisfied and unsatisfied patients regarding telehealth services.

Figure 4 ROC curve.

Figure 5 presents the confusion matrix for the logistic regression model predicting patient satisfaction with telehealth services. The matrix illustrates the model’s performance by showing the number of true positive (TP), true negative (TN), false positive (FP), and false negative (FN) predictions. The top-left cell shows that 626 patients who were satisfied with telehealth were correctly predicted as satisfied (TP). The bottom-right cell indicates that 337 unsatisfied patients were correctly predicted as unsatisfied (TN). The top-right cell shows that 47 satisfied patients were incorrectly predicted as unsatisfied (FP), while the bottom-left cell indicates that 60 unsatisfied patients were incorrectly predicted as satisfied (FN).

Figure 5 Confusion matrix as a plot.

The overall accuracy of 90% indicates that the model performs well in predicting both satisfaction and unsatisfaction. The model is more accurate in predicting satisfaction (93.016%) compared to unsatisfaction (84.887%). There are more false negatives (60) than false positives (47), indicating that the model occasionally misclassifies satisfied instances as unsatisfied. The confusion matrix plot and the accuracy metrics together provide a comprehensive overview of the model’s performance.

Discussion

Telehealth has been a widely used technology for delivery of health care services to patients with chronic diseases. It increases the flexibility and suitability access of health care and decreases health care cost while improving patients’ outcomes. Despite wide use of telehealth, patients’ view on telehealth remains unspecified (Hailu et al., 2024; Fleischhacker, 2020; Ma et al., 2022).

Our study measured patient satisfaction with telehealth services and highlighted factors affecting chronic patient satisfaction with telehealth. We found high overall satisfaction scores among chronic patients, with 62.9% satisfied with the services, while 37.1% were unsatisfied. This indicates that a majority of chronic patients appreciate the benefits and convenience of telehealth, yet a significant minority remain unsatisfied. These results may be related to the ease of scheduling appointments and enhanced communication with healthcare providers through telehealth platforms, which can contribute to higher satisfaction. Additionally, lower costs compared to in-person visits are particularly beneficial for patients managing chronic conditions that require frequent check-ups.

The current results were similar to a cross-sectional survey by Polinski et al. (2016), which investigated patients’ satisfaction and preferences for telehealth visits. They reported that 54% of patients were satisfied with telehealth visits. These findings were supported by Abdulwahab & Zedan (2021), who examined factors affecting patient satisfaction with telemedicine services in outpatient clinics using a cross-sectional survey. They found that 55.7% of patients were satisfied with the services, while only 8.9% were unsatisfied. In addition, a cross-sectional survey by Blavin (2023), revealed that more than half of patients with chronic conditions used telehealth services, most of whom reported positive satisfaction with their telehealth services. Findings indicated that telehealth was appropriate for diverse chronic health conditions. The current study findings were supported by Parsonson et al. (2021), a cross-sectional study that measured patient satisfaction with telehealth consultations in medical oncology clinics during the COVID-19 pandemic. This study revealed that 95% of patients were satisfied with telehealth services and 82% of patients preferred to continue using telehealth services.

The logistic regression results in Table 2 highlight several significant factors associated with chronic patient satisfaction related to telehealth services. The positive and significant constant term (coefficient = 4.129, p < 0.001) indicates a generally high baseline satisfaction level among patients using telehealth services. This suggests that, overall, patients find telehealth services beneficial.

The negative impact of age on satisfaction (coefficient = −0.191, p < 0.001) reveals that older patients are less satisfied with telehealth services. This could be due to difficulties in adapting to new technologies or a preference for face-to-face consultations. Addressing the technological barriers for older adults through user-friendly interfaces and dedicated support could improve their satisfaction levels. Also, the positive association with male sex (coefficient = 0.473, p = 0.047) suggests that male patients are more satisfied with telehealth services compared to female patients. This sex disparity might be due to differences in health service utilization patterns or varying expectations from telehealth services. The strong correlation between having a bachelor’s degree and increased satisfaction (coefficient = 1.977, p < 0.001) indicates that higher educational attainment is associated with better experiences with telehealth. Educated patients may have better digital literacy, making them more comfortable and efficient in using telehealth platforms. Employment status (working) showed a positive association with satisfaction (coefficient = 1.445, p < 0.001). Working individuals might value the convenience and time efficiency offered by telehealth services, which align well with their busy schedules. Having children is positively associated with satisfaction (coefficient = 1.189, p < 0.001). Parents may find telehealth particularly advantageous as it allows them to manage their health needs alongside their responsibilities toward their children more efficiently. Residence far from the hospital is also a significant positive predictor (coefficient = 1.070, p = 0.001). Patients living farther away find telehealth services particularly beneficial as it saves them travel time and effort, making healthcare more accessible.

Also, both using telehealth again and recommending it to others positively influence satisfaction (coefficients = 0.906 and 0.738, p = 0.006 and p = 0.007, respectively). This underscores the importance of positive initial experiences in encouraging repeat use and word-of-mouth promotion of telehealth services. Ease of contact with IT support at the hospital is a significant positive predictor (coefficient = 0.906, p < 0.001). This highlights the critical role of technical support in enhancing patient satisfaction by resolving issues promptly and effectively. Continuous training about using telehealth is a significant positive predictor (coefficient = 1.328, p < 0.001). Providing ongoing education and training for patients ensures they are comfortable and proficient in using telehealth services, leading to higher satisfaction.

These findings were supported by Almalki et al. (2023), who examined other factors affecting patients’ satisfaction, such as the stability of internet connection and easy access to services. A cross-sectional study in Saudi Arabia highlighted a significant disparity in the utilization of telemedicine services across different populations, primarily due to demographic and socioeconomic factors (Almalki et al., 2023). Other studies have shown that patients with higher education levels are more likely to use telemedicine (Stamenova et al., 2020; Yoon et al., 2024). The current study findings were in line with (Almalki et al., 2023), who found that female sex (adjusted odds ratio (AOR) = 2.519), having a higher education level (AOR = 3.434) for secondary education and AOR = 5.87 for higher education), and living in urban areas (AOR = 2.721) were associated with higher odds of telemedicine use. A recent systematic review by Aashima Nanda & Sharma (2021), on patient satisfaction and experience with telemedicine revealed that telemedicine services showed satisfactory levels on various outcome factors, such as patients’ concerns.

Our study found that older patients tend to report lower satisfaction with telehealth services, with higher satisfaction levels observed among male patients. Additionally, a higher level of education, particularly having a bachelor’s degree, was strongly associated with increased satisfaction. These findings contrast with those of Müller, Alstadhaug & Bekkelund (2017), who reported that females were more satisfied with telehealth services than males (p-value 0.027). However, in their study, neither age nor education level was found to significantly affect patient satisfaction.

Conclusion

The findings of this study highlight several key factors influencing patient satisfaction with telehealth services. Older age was associated with lower satisfaction, while male patients and those with higher education levels reported greater satisfaction. Employment status, having children, and willingness to use telehealth again were also positively correlated with satisfaction. Factors such as ease of contact with IT support, living far from the hospital, and receiving continuous training on using telehealth further enhanced satisfaction. These results suggest that telehealth services can be improved by focusing on these demographic and service-related factors to better meet the needs of patients with chronic diseases.

Limitation of study

This study provides valuable insights into factors influencing patient satisfaction with telehealth services, but several limitations warrant discussion. First, the use of a cross-sectional design limits the ability to infer causality between predictors and satisfaction outcomes. Longitudinal studies would be beneficial to explore changes in satisfaction over time and the impact of sustained telehealth use.

Second, the reliance on self-reported data introduces the possibility of response bias, as participants may overestimate or underestimate their satisfaction levels due to subjective perceptions. Future studies could incorporate objective measures, such as system usage data, to validate self-reported satisfaction scores.

Third, the study was conducted in private hospitals, potentially limiting generalizability to public healthcare settings and rural populations, where telehealth adoption and challenges may differ. Expanding the research to include diverse healthcare environments would enhance applicability and provide a more comprehensive understanding.

Recommendations

Future studies will aim to incorporate additional variables, such as prior healthcare experiences, comorbidities, and digital literacy, to further reduce the risk of confounding and improve model validity.

Provide additional tech support and personalized care options. Provide continuous telehealth usage training, especially for older patients.

Implications of the study

The findings imply that telehealth services can be optimized by addressing the specific needs and preferences of different patient demographics. Understanding that older patients might be less satisfied can help in designing more age-friendly telehealth interfaces. The positive correlation with education level and employment status suggests that telehealth services may be particularly effective for educated and employed individuals. Continuous training and ease of IT contact enhance satisfaction, indicating that support and education are vital components of successful telehealth implementation. Future studies should aim to include a broader range of chronic diseases to improve the generalizability of the findings.

Strengths of the study

1. The study included 1,070 participants, enhancing data robustness and generalizability.

2. Multiple demographic, socioeconomic, and telehealth-related factors were analysed, providing a holistic view of patient satisfaction.

3. Focused on patients with chronic diseases, which is highly relevant to current telehealth applications in healthcare.

4. Used logistic regression to identify key predictors of satisfaction, providing statistical reliability and insight into the factors influencing satisfaction.

Supplemental Information

Supplemental Information 1 Raw data

Supplemental Information 2 Questionnaire

Additional Information and Declarations

Competing Interests

Author Contributions

Human Ethics

Data Availability

The authors declare there are no competing interests.

Abdelaziz Hendy conceived and designed the experiments, performed the experiments, prepared figures and/or tables, authored or reviewed drafts of the article, and approved the final draft.

Sally Mohammed Farghaly Abdelaliem analyzed the data, authored or reviewed drafts of the article, editing, and approved the final draft.

Ahmed Zaher conceived and designed the experiments, performed the experiments, prepared figures and/or tables, and approved the final draft.

Bothayna N. Sadek conceived and designed the experiments, performed the experiments, authored or reviewed drafts of the article, and approved the final draft.

Abdulqadir J. Nashwan conceived and designed the experiments, prepared figures and/or tables, authored or reviewed drafts of the article, and approved the final draft.

Mohammed Musaed Ahmed Al-Jabri conceived and designed the experiments, prepared figures and/or tables, authored or reviewed drafts of the article, and approved the final draft.

Ahmad Ahmeda performed the experiments, analyzed the data, prepared figures and/or tables, and approved the final draft.

Ahmed Hendy performed the experiments, analyzed the data, prepared figures and/or tables, authored or reviewed drafts of the article, editing, and approved the final draft.

Amany Anwar Saeed Alabdullah conceived and designed the experiments, analyzed the data, authored or reviewed drafts of the article, and approved the final draft.

Shaban Majed Sinnokrot performed the experiments, prepared figures and/or tables, authored or reviewed drafts of the article, and approved the final draft.

The following information was supplied relating to ethical approvals (i.e., approving body and any reference numbers):

Ethical approval to conduct the research was obtained from the Research Ethical Committee at the Faculty of Medicine, MTI University ID 1023-2024.

The following information was supplied regarding data availability:

The raw data are available in the Supplementary File.

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
