# Peer review of "Telehealth satisfaction among patients with chronic diseases: a cross-sectional analysis"

_PeerJ, doi:10.7717/peerj.19245_

## Round 0.1 · original submission · Major Revisions

The reviewers have provided extensive comments. Please address them in a thorough revision.

In particular the reviewers have highlighted issues with your statistical analyses

Note: Reviewer 1 has provided an annotated PDF

Reviewer 1 ·

Basic reporting

Interesting paper and relevant to modern health care.

Experimental design

a few methodological issues-
The setting of the study in private health care needs to be made clear,
Please provide information on how representative the sample is of the usual care population within the health system, this will help inform generalisability of findings.

Validity of the findings

the model needs to be tested for confounding by including variables linked with outcome variable.
Model performance may be erroneous.
Sample calculation needs clarification with regards to key outcome

Additional comments

Please discuss limitations and suggestions for future research.
Comment included in attached document

Annotated reviews are not available for download in order to protect the identity of reviewers who chose to remain anonymous.

Reviewer 2 ·

Basic reporting

I appreciate the research team for the topic you chose.The research language is clear and understandable. Some concerns need to be addressed.

The introduction and background need to be expanded with support from the literature. Consider clarifying in more detail which telehealth components/approaches you are addressing for a better overview.
Were the telehealth components used in your country similar to the telehealth applications in the literature you provided in the introduction?

Experimental design

You should state why you defined your study as cross-sectional. What criteria did you use to determine that this was a prevalence study?
How diverse was the sample in terms of chronic diseases? Providing diversity in the sample may increase its generalizability, otherwise, this should be added as a limitation.

The sample calculation is clearly explained, but a sample size more than twice the calculated value was selected. Why did you need this?
This situation carries the risk of affecting the clinical significance of the study. In this case, is the confusion matrix for the logistic regression model still valid if it is kept above the minimum number calculated in the sample?

You should state the suitability of the scale used for the Arab society in terms of validity and reliability, otherwise this situation creates a significant concern. Even if the Cronbach's Alpha value is good, this alone is not sufficient. How was language translation provided? Was expert opinion obtained, how many experts? It was stated that content validity was performed, which technique was used for this? How was construct validity assessed? Was Confirmatory Factor Analysis applied?
The sample number in Table 1 does not match the findings.

Validity of the findings

The results are clearly stated.
Is the confusion matrix for the logistic regression model still valid data if the calculated minimum number in the sample is kept above?

Which chronic diseases did you include in your study? Consider adding similarities and differences in the literature you discussed.

In the later sections of the discussion, examples of existing studies are presented, contrary to what is stated in the first paragraph. You should clearly state the difference in your study.

Additional comments

I hope that my comments will further reveal the effort you are putting into your current work.

Reviewer 3 ·

Basic reporting

1) in the result section of the abstract, you should add numbers (%), not plain description 

2) the first time you should write the full name, e.g., e.g (IT, AUC ) in the abstract 


3) in the title (A Comprehensive Study) -------while in the abstract descriptive cross-sectional --- resolve and fix 


4) the objective should be smart (time, person, place) in the abstract and the manuscript 

5)  what is the difference between both? 

To analyze the impact of demographic factors (age, gender, marital status) on patient satisfaction
130    with telehealth.
 To evaluate the influence of socio-economic variables (education level, income level,    employment status) on patient satisfaction with telehealth.
 
6) The aim and the objective seem to be similar --- the aim should be one broad nonspecific, while the objective should be smart (time, person, place)

7) Patients must suffer from a chronic disease but not be in a critical condition. What do you mean by critical conditions? 



13) Technology data of patients--- What do you mean by it? 

14) The term gender refers to behavior type; substitute it with sex.

15)196 A pilot study involving 110 patients (10% of the total sample size) was conducted to assess the
197    clarity, applicability, relevance, and feasibility of the survey tools and to estimate the data
198    collection duration. No modifications were made based on the pilot study results, and the patients
199    involved in the pilot were included in the main study sample. ------should move to reliability and validity. 


16) Many grammar and editing errors, e.g., p. italic/lline 92

1`7) Add about the prevalence of chronic disease in Egypt.
18) Please provide a definition of chronic disease.
19) Add more details about the evaluated service (history, who provided the procedures) in the introduction, not about the telehealth in Egypt. 

19) in the results demographic profile of the 1217 studied participants---------while in the Consequently, data were collected from 1305 patients. During data cleaning and revision, 235 entries were excluded due to missing data, resulting in a final analyzed sample size of 1070 patients, Figure 1.

Experimental design

8) 140 A purposive sample of chronic patients who used telehealth for follow-up with their physician in the  study setting was selected. ----- What is the selected study setting? 


9) This questionnaire was developed based on an extensive review of relevant studies and previously published instruments (23) ------ studies/study 23 

10) by a panel of five experts who evaluated its representativeness for the target construct. Who is this expert???


11)189 Fieldwork:----- should be data collection process

12) . The first section outlined the  study objectives and eligibility criteria. The second section gathered demographic, socioeconomic, and  technology-related information, as  well as  the Telehealth Satisfaction Scale. While in another area. ---170 Tools of Data Collection ----------------------178 Part I: The first part of the questionnaire gathered the demographic, socioeconomic, and technological
179 data points of patients, including age, gender, education level, marital status, employment status, and income
180 level, have children, use telehealth again, recommend telehealth to others, easy to contact with it at
181 hospital, residence, continuous training about using telehealth.
182 Part II: Telehealth Satisfaction Scale (TeSS)--------fix and correct 


20) the conclusion should be extracted from the results ? rewrite please

Validity of the findings

before the regression you should make an association




add the mean and SD of the TeSS --- in association with the demographic data


what is the recommendation ?

What is the strength of this study ?

---

## Round 0.2 · Major Revisions

An additional round of revisions is required. The comments from both reviewers must be addressed or the submission may be rejected

Reviewer 1 ·

Basic reporting

The authors have made a number of significant changes to the document that have clarified the background for the study, the setting and design.

The document reads better overall.

Experimental design

satisfactory

Validity of the findings

I remain concerned that the model uses variables that are likely to be co-linear with output variable as it uses patient satisfaction as one of the model inputs. Please can the authors calculate model performance without

Use Telehealth Again (Yes)
Recommend Telehealth to Others (Yes)

Additional comments

none

Reviewer 2 ·

Basic reporting

"Yes, telehealth components used in the study setting similar to the telehealth applications in the literature provided in the introduction" should be explained more clearly (perhaps with national resources)
In the introduction, it would be good to explain which telehealth components are used in your country. Or I suggest you address which telehealth component you are evaluating in the method. Otherwise telehealth is a term that includes very broad concepts, as you mentioned.

Experimental design

"Yes, telehealth components used in the study setting similar to the telehealth applications in the literature provided in the introduction" should be explained more clearly (perhaps with national resources)
In the introduction, it would be good to explain which telehealth components are used in your country. Or I suggest you address which telehealth component you are evaluating in the method. Otherwise, telehealth is a term that includes very broad concepts, as you mentioned.


In testing the suitability of the Telehealth Satisfaction Scale (TeSS) for your culture, it is important to include the Kaiser-Meyer Olkin-Bartlett test, exploratory and confirmatory factor analysis results to test the construct validity.

Validity of the findings

The selection of sample size does not necessarily mean that the larger, the better. The fact that the current study has reached a larger sample size than necessary makes the increase in the accuracy of the study results small and clinically insignificant after a certain point and raises ethical issues. The excess requires a literature-supported explanation.

Additional comments

The referee's responses must be given directly to the questions.The referee should not search from the manuscript

---

## Round 0.3 · accepted · Accept

Manuscript revisions assessed, ready for publication.

Reviewer 1 ·

Basic reporting

happy with changes.
document reads better
R2 more realistic

Experimental design

happy with changes.
document reads better
R2 more realistic

Validity of the findings

happy with changes.
document reads better
R2 more realistic

Additional comments

happy with changes.
document reads better
R2 more realistic

Reviewer 2 ·

Basic reporting

The document reads better overall. I appreciate the efforts of the authors. The manuscript is suitable in its current state.

Experimental design

The document reads better overall. I appreciate the efforts of the authors. The manuscript is suitable in its current state.

Validity of the findings

The document reads better overall. I appreciate the efforts of the authors. The manuscript is suitable in its current state.